# Inhibition of Adipose Tissue Beiging by HIV Integrase Inhibitors, Dolutegravir and Bictegravir, Is Associated with Adipocyte Hypertrophy, Hypoxia, Elevated Fibrosis, and Insulin Resistance in Simian Adipose Tissue and Human Adipocytes

**DOI:** 10.3390/cells11111841

**Published:** 2022-06-04

**Authors:** Kenza Ngono Ayissi, Jennifer Gorwood, Laura Le Pelletier, Christine Bourgeois, Carine Beaupère, Martine Auclair, Roberta Foresti, Roberto Motterlini, Michael Atlan, Aurélie Barrail-Tran, Roger Le Grand, Delphine Desjardins, Bruno Fève, Olivier Lambotte, Jacqueline Capeau, Véronique Béréziat, Claire Lagathu

**Affiliations:** 1Inserm UMR_S938, Centre de Recherche Saint-Antoine, Institut Hospitalo-Universitaire de Cardio-Métabolisme et Nutrition (ICAN), Sorbonne Université, 75012 Paris, France; kenza.ngono@inserm.fr (K.N.A.); jennifergorwood@yahoo.fr (J.G.); laura.le-pelletier@inserm.fr (L.L.P.); carine.beaupere@inserm.fr (C.B.); martine.auclair@inserm.fr (M.A.); drmichaelatlan@gmail.com (M.A.); bruno.feve@inserm.fr (B.F.); jacqueline.capeau@inserm.fr (J.C.); 2UMR1184 Inserm, Center for Immunology of Viral Infections and Autoimmune Diseases, IDMIT Department, IBFJ, CEA, Université Paris Saclay, 92032 Fontenay-aux-Roses, France; christine.bourgeois@universite-paris-saclay.fr (C.B.); aurelie.barrail-tran@bct.aphp.fr (A.B.-T.); roger.legrand@cea.fr (R.L.G.); delphine.desjardins@cea.fr (D.D.); olivier.lambotte@aphp.fr (O.L.); 3INSERM UMR_S955, IMRB, Université Paris-Est Créteil, 94000 Créteil, France; roberta.foresti@inserm.fr (R.F.); roberto.motterlini@inserm.fr (R.M.); 4Service de Chirurgie Plastique et Esthétique, Hôpital Tenon, AP-HP, 75020 Paris, France; 5Service d’Endocrinologie, Diabétologie et Reproduction, Hôpital Saint-Antoine, CRMR, PRISIS, AP-HP, 75012 Paris, France; 6Service de Médecine Interne et Immunologie Clinique, Hôpital Bicêtre, AP-HP, 94270 Kremlin-Bicêtre, France

**Keywords:** HIV integrase inhibitors, adipose tissue, beiging, fibrosis, hypertrophy, insulin resistance

## Abstract

For people living with HIV, treatment with integrase-strand-transfer-inhibitors (INSTIs) can promote adipose tissue (AT) gain. We previously demonstrated that INSTIs can induce hypertrophy and fibrosis in AT of macaques and humans. By promoting energy expenditure, the emergence of beige adipocytes in white AT (beiging) could play an important role by limiting excess lipid storage and associated adipocyte dysfunction. We hypothesized that INSTIs could alter AT via beiging inhibition. Fibrosis and gene expression were measured in subcutaneous (SCAT) and visceral AT (VAT) from SIV-infected, dolutegravir-treated (SIVART) macaques. Beiging capacity was assessed in human adipose stromal cells (ASCs) undergoing differentiation and being exposed to dolutegravir, bictegravir, or raltegravir. Expression of beige markers, such as positive-regulatory-domain-containing-16 (PRDM16), were lower in AT of SIVART as compared to control macaques, whereas fibrosis-related genes were higher. Dolutegravir and bictegravir inhibited beige differentiation in ASCs, as shown by lower expression of beige markers and lower cell respiration. INSTIs also induced a hypertrophic insulin-resistant state associated with a pro-fibrotic phenotype. Our results indicate that adipocyte hypertrophy induced by INSTIs is involved via hypoxia (revealed by a greater hypoxia-inducible-factor-1-alpha gene expression) in fat fibrosis, beiging inhibition, and thus (via positive feedback), probably, further hypertrophy and associated insulin resistance.

## 1. Introduction

The importance of adipose tissue (AT) as a simian immunodeficiency virus (SIV)/human immunodeficiency virus (HIV) reservoir has been previously outlined, and we have identified AT as a crucial cofactor in both viral persistence and chronic immune activation/inflammation during HIV infection [1,2]. Moreover, we revealed that HIV/SIV infection per se increases fat fibrosis and alters adipogenesis, probably through the release of viral proteins [3] and that HIV/SIV promotes AT senescence, which in turn may alter adipocyte function and contribute to insulin resistance [4].

Recently, weight gain and fat expansion were observed in some HIV-infected antiretroviral therapy (ART)-controlled people living with HIV (PLWH) who switch to the new ART class of integrase-strand-transfer-inhibitors (INSTIs), such as dolutegravir and bictegravir [5,6,7,8]. Among PLWH, overweight and obesity are of concern, given the worldwide increase in the prevalence of obesity. Indeed, weight gain is associated with a higher risk of cardiovascular and metabolic diseases [9,10]. Risk factors for this weight gain include age, ethnicity, body mass index (BMI), and female sex [11]. However, our present understanding of the mechanisms involved remain unknown.

To maintain energy homeostasis, healthy AT has a high level of plasticity with remodeling capacities of extracellular matrix (ECM) [12,13]. In response to an excess energy supply, AT expansion leads to adipocyte hypertrophy (increase in size of existing adipocytes) and, in some cases, adipocyte hyperplasia (increase in the number of adipocytes recruited from adipose stromal cells) associated with fibrosis (with the accumulation of collagens 1 and 6) and hypoxia (with the induction of hypoxia-inducible-factor-1-alpha (HIF-1α)), leading then to AT dysfunction and progression towards insulin resistance [12,14]. We previously demonstrated an unhealthy AT expansion characterized by adipocyte hypertrophy, fibrosis, and insulin resistance in response to INSTI treatment [15].

Whereas WAT is the most abundant AT depot in the body (storing and releasing lipids, depending on physiological needs), brown AT is a minor depot in adult humans with thermogenic capacities mediated primarily by uncoupling protein 1 (UCP1), which dissipates the mitochondrial proton gradient generated by lipid catabolism and thus, produces heat [16]. More recently, beige adipocytes with certain properties of brown adipocytes have been discovered in WAT depots. These adipocytes could arise from either de novo beige differentiation of adipose stromal cells (ASCs) or trans-differentiation of preexisting white adipocytes [17] By increasing fatty acid oxidation and reducing adipocyte hypertrophy, greater WAT beiging limits the fat gain [17,18].

There is a reciprocal negative relationship between AT fibrosis and beiging. Indeed, transforming growth factor beta (TGF-β), HIF-1α, and discoidin domain receptor tyrosine kinase 1 (DDR1) have been shown to promote fibrosis and inhibit AT beiging in response to adipocyte hypertrophy [19,20,21,22]. Conversely, beige adipogenesis represses fibrosis [20,21,23]. We hypothesized that INSTI-induced adipocyte hypertrophy and fibrosis are related to the inhibition of WAT beiging capacity.

Here, we characterized the impact of dolutegravir on adipogenesis, beiging, and fibrosis in subcutaneous (SCAT) and visceral AT (VAT) from SIV-infected, ART-controlled macaques (SIVART). In vitro, we differentiated human ASCs into beige or white adipocytes and evaluated the impacts of INSTIs on adipogenesis, hypoxia, lipid accumulation, respiration, and ECM production. We found that INSTI-induced adipocyte hypertrophy and increased fibrosis were linked (through hypoxia) to the inhibition of fat beiging.

## 2. Materials and Methods

### 2.1. Adipose Tissue Samples from Macaques

Cynomolgus macaques (Macaca fascicularis) were housed in the CEA animal facility (IDMIT (Fontenay-aux-Roses, France); government accreditation: D92-032-02). Eleven SIV-infected macaques from the ANRS SIVART cohort (mean ± standard error of the mean (SEM) age: 9.40 ± 0.14 years; weight: 9.29 ± 0.37 kg), had been treated orally and by subcutaneous injection with tenofovir-disoproxil-fumarate (TDF, 20 mg/kg), emtricitabine (FTC, 50 mg/kg), and dolutegravir (DTG, 20 mg/kg) for 2.18 ± 0.03 years and were compared with nine control macaques, which were uninfected and untreated (mean age: 6.78 ± 0.92 years; mean weight: 9.07 ± 0.98 kg). All SIVART macaques under antiretroviral treatment had undetectable viral loads (<50 copies/mL). The study was approved by the French Ministry of Education (CEEA 44; reference 2015102713323361.02, APAFIS#2453). The animal facility complied with the Standards of the U.S. Office for Laboratory Animal Welfare (#A5826-01) and the European Directive 2010/63/EU (recommendation #9). At necropsy, SCAT and VAT samples were collected, fixed in 4% paraformaldehyde (Merck, Sigma-Aldrich, St Louis, MO, USA), and stained with Sirius red. Fibrosis index (% of total surface) was performed using CaloPix software (Tribvn, Chatillon, France).

### 2.2. Isolation and Differentiation of ASCs

Human SCAT samples were obtained from nine healthy women (age 42.7 ± 2.9 years; BMI = 25 ± 1.35 kg/m^2^). All provided their prior, written, informed consent. The research complied with the tenets of the Declaration of Helsinki and was approved by an independent ethics committee. ASCs were isolated, as described previously [3]. Beige adipocyte differentiation of ASCs was induced by the addition of proadipogenic Dulbecco Modified Eagle’s Medium (DMEM), 4.5 g/L glucose (Thermo Fisher Scientific, Waltham, MA, USA) containing 10% FBS, 2 mmol/L glutamine, 100 U/mL penicillin/streptomycin, 10 mmol/L HEPES, 1 µmol/L dexamethasone, 500 µmol/L 3-Isobutyl1-methylxanthine (IBMX), 1 µmol/L rosiglitazone, 1 µmol/L insulin, 50 μmol/L indomethacin, and 2 nmol/L T3 (Merck, Sigma-Aldrich) for seven days and then maintained in DMEM with rosiglitazone, insulin, and T3 up until day 14 [24] (See Appendix A). White differentiation was induced for 14 days, as described previously [3]. Throughout the differentiation period, ASCs were exposed or not exposed to concentrations of dolutegravir, raltegravir, or bictegravir (Selleck Chemicals, Houston, TX, USA) near the Cmax observed in PLWH (dolutegravir: 3.1 μg/mL, raltegravir: 2.1 μg/mL; bictegravir: 6.2 μg/mL) or to 0.1% dimethyl-sulfoxide (DMSO, Merck, Sigma-Aldrich) as a control. In some experiments, ASCs were exposed simultaneously to INSTIs and to 5 µmol/L PX478 (Adooq Bioscience, Irvine, CA, USA).

### 2.3. Protein Extraction and Western Blotting

Proteins were extracted from cell monolayers, separated using SDS-PAGE, blotted onto nitrocellulose membranes [15], and detected using specific antibodies against peroxisome-proliferator-activated receptor γ (PPAR γ), PPARγ-coactivator-1α (PGC-1α) (Cell Signaling Technology, Danvers, MA, USA), UCP1 (Thermo Fisher Scientific), collagen-1-α1, collagen-6-α1 (SCBT, Dallas, TX, USA), HIF-1α (Novus Biologicals, Littleton, CO, USA), and tubulin (Merck, Sigma-Aldrich). The signal was measured with the iBright Imaging Systems (ThermoFisher Scientific). Beige adipocytes were serum-starved for 18 h. In a Western blot assay, insulin sensitivity was evaluated after 7 min of exposure to 100 nM insulin as the ratio between phosphorylated Akt (phospho-Akt-ser473) and total Akt (Cell Signaling Technology).

### 2.4. Protein Secretion into the Cell Culture Media

The levels of human FGF21 secreted by differentiated ASCs after 14 days of differentiation were determined in the culture medium from the last 48 h using Quantikine ELISA kit and according to the manufacturer’s instructions (Bio-Techne, Minneapolis, MN, USA).

### 2.5. RNA Isolation and Quantitative PCR

Total RNA from macaque AT samples was isolated using QIAzol reagent (Qiagen, Courtaboeuf, France). Total RNA was isolated from cultured cells using RNeasy mini-columns (Qiagen). mRNA expression was analyzed using PCR [15] (see Appendix A for oligonucleotide sequences). PCR raw data were analyzed using the relative quantification approach with efficiency correction. The internal reference gene, *PPIA*, was measured in addition to the target genes for each sample. Ratios between the gene of interest and the reference gene were evaluated and compared.

### 2.6. Oxidative Stress and Mitochondrial Dysfunctions

The mitochondrial membrane potential was evaluated with the cationic dye tetra-chloro-tetra-ethyl-benzimidazolyl-carbocyanine-iodide (JC1) and the mitochondrial mass was evaluated using MitoTracker (ThermoFisher Scientific) [15]. Cells were incubated with JC-1 or MitoTracker for 2 h at 37 °C. The results were quantified on a plate fluorescence reader. Cells were incubated with 5-6-chloromethyl-2,7-dichlorodihydrofluorescein diacetate (CM-H_2_DCFDA) for 2 h at 37 °C to assess the production of ROS (Invitrogen Corporation). The fluorescence was quantified on a plate reader (Spectrafluor Plus, Tecan, Trappes, France). The results were normalized against 4′,6-diamidino-2-phenylindole (DAPI) fluorescence.

### 2.7. Oxygen Consumption Measurements

The oxygen consumption rates of white and beige adipocytes were assessed using a Seahorse XF24 extracellular flow analyzer (Agilent Technologies, Santa Clara, CA, USA) [25]. Basal OCR was measured for 30 min. Several parameters such as ATP-related respiration, maximal respiration, proton leak, and non-mitochondrial respiration were assessed using the Mito Stress test for a total duration of 120 min by the addition of 1 μg/mL oligomycin, 0.7 μmol/L carbonyl cyanide 4-(trifluoromethoxy) phenylhydrazone (FCCP), and 1 μmol/L rotenone/antimycin A. Data were normalized against the protein content.

### 2.8. Lipid Metabolism

To evaluate lipid accumulation, cells were stained with Oil-Red-O (Merck, Sigma-Aldrich) [20]. Lipolysis was assessed using the Glycerol-Glo™ Assay kit (Promega, Madison, WI, USA) after 120 min of incubation with 1 µM isoproterenol. The luminescence (Spectrafluor Plus, Tecan, Trappes, France) was normalized against the protein content.

### 2.9. Statistical Analysis

Differences between SIVART and control macaques and between INSTI-treated and control cells were evaluated using the non-parametric Mann–Whitney test. Correlations between gene expression data were analyzed by calculating Pearson’s coefficient since the relationships exhibited linearity. The threshold for statistical significance was set to *p* < 0.05. We performed a two-way ANOVA to study the impact of PX478 on gene expression in ASCs. All analyses were performed using Prism 5.0 software (GraphPad Software, La Jolla, CA, USA).

## 3. Results

### 3.1. SCAT and VAT from SIV-Infected Dolutegravir-Treated Macaques Display Higher Levels of PPARG and Profibrotic Gene Expression but Lower Levels of Beige Marker Gene Expression

The mRNA expression level of PPARG was higher in SCAT and VAT from SIVART macaques than in samples from controls (Figure 1A). We previously reported elevated PPARG expression in AT from uninfected, DTG/TDF/FTC-treated macaques exhibiting adipocyte hypertrophy [15]. The mRNA expression levels of the beige-specific markers, transmembrane protein 26 (TMEM26) and positive-regulatory domain containing 16 (PRDM16) were lower in SCAT and VAT from SIVART than in samples from controls (Figure 1B). For SIVART versus control animals, the fibrosis index was respectively 25.5% versus 17.8% in SCAT (*p* = 0.048) and 27.7% versus 14.9% in VAT (*p* = 0.025) (Figure 1C–D). Accordingly, the mRNA expression levels of the fibrosis-related genes, COL1A2, COL6A1, TGFB1, and fibronectin (FN), were higher in SCAT and VAT from SIVART animals versus controls (Figure 1E). Expression of the beige marker, PRDM16, was negatively correlated with the expression of fibrosis-related genes, COL1A2 (r = −0.702; *p* = 0.017) and FN (r = −0.628; *p* = 0.026) in SCAT and with the fibrosis index in VAT (r = −0.512; *p* = 0.027). These data indicate that INSTI-induced beiging inhibition was related to elevated AT fibrosis.

### 3.2. INSTIs Are Associated with Elevated Lipogenic Gene Expression and Insulin Resistance in Beige Adipocytes

We next evaluated the respective impacts of dolutegravir, raltegravir, and bictegravir on ASCs differentiating into beige adipocytes. INSTI treatment was associated with greater lipid accumulation than in the control experiment using the vehicle (DMSO) (Figure 2A); in line with our in vivo data, we observed elevated mRNA and protein expression of the adipogenic marker, PPARγ (Figure 2B). Dolutegravir, bictegravir and (albeit to a lesser extent) raltegravir induced greater mRNA expression of the pro-lipogenic factor sterol-regulatory-element-binding-factor-1c (SREBF1C) and the lipid metabolism markers fatty acid synthase (FASN), stearoyl-CoA-desaturase-1 (SCD1), and fatty-acid-binding-protein-4 (FABP4) (Figure 2C). Despite the INSTIs’ pro-adipogenic and pro-lipogenic effects, these drugs promoted oxidative stress (Figure 2D) and insulin resistance. Indeed, both dolutegravir and bictegravir inhibited acute insulin-induced phosphorylation of Akt (a key enzyme of the insulin-signaling pathway) (Figure 2E). In addition, INSTIs were associated with a lower level of basal and β-adrenergic-stimulated lipolysis (Figure 2F).

### 3.3. Dolutegravir and Bictegravir Inhibit Beige Differentiation

We evaluated beige marker levels in white- and beige-differentiating adipocytes. Dolutegravir or bictegravir reduced the expression of the beige-specific markers, TMEM26, cluster-of-differentiation-137 (CD137), Cbp/P300-interacting-transactivator-1 (CITED1), PRDM16, and iodothyronine-deiodinase-2 (DIO2) (Figure 3A). Dolutegravir and bictegravir were associated with lower mRNA and protein expression of UCP1 (Figure 3B) and the brown/beige-specific adipokine FGF21 (Figure 3C). It is noteworthy that the levels of expression of all beige markers in dolutegravir/bictegravir-treated cells were similar to those observed in white adipocytes, suggesting a whitening in response to INSTIs.

### 3.4. Dolutegravir and Bictegravir Are Associated with Impaired Mitochondrial Integrity in Beige Adipocytes

In line with the observed inhibition of beige differentiation, INSTI treatment reduced mitochondrial function and respiration. Dolutegravir and bictegravir were associated with lower protein expression of PGC-1α (an inducer of mitochondrial biogenesis) (Figure 4A), a lower mitochondrial membrane potential (Figure 4B), a lower oxygen consumption rate (Figure 4C), and lower basal and maximal mitochondrial respiration rates (Figure 4D). Proton leakage and uncoupled mitochondrial respiration rates were similar to those observed in white adipocytes (Figure 4D).

### 3.5. Dolutegravir Enhances the Expression of Profibrotic Genes in Beige Adipocytes

In line with our in vivo data, dolutegravir increased the expression of collagens (COL1A2, COL6A1 and COL1A1) (Figure 5A,B), lysyl-oxidase (LOX, a crucial enzyme for collagen fiber crosslinking), the myofibroblast marker alpha smooth muscle actin (ACTA2), and the profibrotic mediators TGF-β (TGFB1) and DDR1 (Figure 5A).

### 3.6. Treatment with Dolutegravir Is Associated with Elevated Expression of HIF1A, Which Is Causally Related to Beiging Inhibition and Fibrosis In Vivo and In Vitro

*HIF1A* levels were elevated in SCAT and VAT from SIVART macaques (Figure 6A). Given that the hypoxia caused by adipocyte hypertrophy can induce AT fibrosis and suppress brown/beige adipogenesis [19], we looked for relationships between these processes. We observed in vivo that *HIF1A* expression was (i) positively correlated with PPARG expression and negatively correlated with expression of the beige marker, PRDM16, in SCAT, and (ii) positively correlated with expression of the fibrotic markers COL6A1, COL6A2, and TGFB1 in VAT (Table 1 and Appendix A). Similarly, dolutegravir treatment in vitro was associated with elevated *HIF1A* mRNA and HIF-1α protein levels (Figure 6B). *HIF1A* mRNA levels were correlated with markers of adipocyte lipid content (FABP4 and Oil-Red-O staining) and mRNA expression of DDR1 (associated with adipocyte hypertrophy [22]) (Table 2). *HIF1A* expression was negatively correlated with the expression of beige markers, PRDM16, UCP1, and DIO2, and positively correlated with the expression of fibrosis markers, LOX, COL1A2, and ACTA2 (Table 2).

To further assess the involvement of HIF-1α in INSTI-induced beiging inhibition and fibrosis, ASCs were concomitantly treated with INSTIs and PX478 (HIF-1α inhibitor) throughout the differentiation process. PX478 suppressed the increase in *HIF1A* expression induced by dolutegravir and bictegravir (Figure 7A). Accordingly, PX478-treatment prevented the reduction in beige markers (Figure 7B) and reduced the enhanced expression of the fibrosis-related gene, LOX, induced by the two drugs (Figure 7C). These results directly argue for a causative role for hypoxia (probably resulting from adipocyte hypertrophy) in the INSTI’s harmful effects (i.e., beiging inhibition and enhanced fibrosis).

## 4. Discussion

To identify the mechanisms underlying AT gain and dysfunctions in PLWH receiving an INSTI-containing drug regimen, we used a simian model of long-term SIV-infected and DTG/FTC/TDF-treated macaques (SIVART), close to the situation of ART-controlled PLWH. As observed previously in non-SIV-infected macaques [15], we confirmed that dolutegravir treatment resulted in higher adipogenesis and fibrosis (See Appendix A). Moreover, we found that the dolutegravir-induced SCAT and VAT adipogenesis and fibrosis were related to the inhibition of WAT-beiging capacity probably via the induction of hypoxia. To ascertain the direct role of INSTI, we used an in vitro model with human ASCs differentiated into beige adipocytes [24]. Dolutegravir- and bictegravir-induced beiging inhibition and fibrosis enhancement in human ASC-derived beige adipocytes were linked to adipocyte lipid accumulation and resulted from increased HIF-1α expression.

For both in vitro and in vivo, we observed elevated *PPARG* expression—confirming that INSTIs favor lipid storage [15]. In vitro, INSTIs were more strongly associated with elevated expression of lipid-metabolism-related genes (*FAS*, *SREBF1C*, *SCD1*, and *FABP4*) than with *PPARG* expression. Moreover, dolutegravir and bictegravir inhibited lipolysis, thus limiting lipid flux. These findings indicate that in addition to elevation of adipogenesis, INSTIs directly promoted lipid accumulation (suggesting adipocyte hypertrophy) together with insulin resistance.

Beige adipocytes play a major role in WAT homeostasis via elevated energy expenditure. We observed low WAT beiging capacity in SCAT and VAT from SIVART macaques. Although the expression of beige-specific genes has already been assessed in SCAT from PLWH with lipodystrophy, the low number of INSTI-treated people prevented a reliable analysis of the drugs’ effect [26]. It has been reported that dolutegravir inhibits brown AT in mice [27] but the WAT’s beiging capacity was not evaluated. Thus, the present study is the first to have revealed dolutegravir’s ability to inhibit beiging in abdominal SCAT and VAT. Moreover, dolutegravir and bictegravir directly impaired beige differentiation and reduced mitochondrial biogenesis and cell respiration in vitro. Overall, our results suggest that both INSTIs may favor adipocyte hypertrophy through beige adipocytes whitening.

It has already been shown that WAT beiging suppresses the AT fibrosis involved in AT dysfunction [23]. We observed greater fibrosis in SCAT and VAT from SIVART macaques, together with elevated expression of fibrosis-related genes (including collagen-6) associated with poor metabolic outcomes [14]. Dolutegravir-treated ASC-derived beige adipocytes exhibited a profibrotic profile. Both in vivo and in vitro, beige markers were negatively correlated with fibrosis-related gene expression; this reinforces the hypothesis of a negative relationship between beiging and fibrosis as a result of INSTI-induced adipocyte hypertrophy. It has been reported that DDR1 exerts a profibrotic role and suppresses beige fat in a model of adipocyte hypertrophy [22]. Accordingly, we found that the elevated *DDR1* expression in dolutegravir-treated beige adipocytes was correlated positively with the expression of several profibrotic genes and correlated negatively with the expression of beiging markers.

Obesity is associated with AT expansion, which can result from either through adipocyte hypertrophy or hyperplasia (formation of new adipocytes from the differentiation of resident adipose stromal cells). In obese AT, hypertrophic expansion of adipocytes is not matched by new vessel formation, creating a hypoxic environment and enhancing HIF-1α expression. In our study, we observed elevated *HIF1A* expression level in SCAT and VAT from SIVART macaques and in beige adipocytes treated with INSTIs. Interestingly, *HIF1A* expression was positively correlated with the expression of *PPARG* in SCAT from SIVART macaques and the expression of *FABP4* in beige adipocytes. Activated HIF-1α can rapidly enhance the expression of profibrotic genes (those coding for LOX and collagens) [14]. Accordingly, we found that *HIF1A* expression was correlated with profibrotic gene expression both in SCAT from SIVART macaques and in INSTI-treated beige adipocytes. HIF-1α can impede beige adipogenesis [28]. Here, we observed that *HIF1A* expression was correlated negatively with the expression of beige markers in SCAT from SIVART macaques and in INSTI-treated beige adipocytes. Importantly, treatment with the HIF-1α inhibitor (PX478) reduced *HIF1A* expression and countered the negative impact of INSTIs on beiging and probably fibrosis; this finding enlightens the causative role of hypoxia in dolutegravir- and bictegravir-induced adverse effects on adipose tissue. Altogether, these data suggest that in response to some INSTIs, HIF-1α is associated with adipocyte hypertrophy and induces fibrosis and impaired beiging capacity.

During the last years, it has been shown that WAT beiging could contribute to carcinogenesis and its related cachexia characterized by massive lipolytic fat degradation [29,30]. Cancer is one of the leading causes of death amongst PLWH. While the incidence of acquired immune deficiency syndrome (AIDS)-defining cancers (ADCs) has significantly decreased in treated PLWH, the incidence of some non-ADCs has increased [31]. In a recent study including almost 30 000 PLWH from the RESPOND cohort and over 1000 cancer events, no overall association between cumulative INSTIs exposure and cancer risk has been observed. The risk of cancer decreased with increasing exposure to INSTIs among ART-naive individuals, which was mainly driven by a decreasing incidence of ADCs [32]. Longer follow-up is needed to confirm this finding.

Our study had a number of limitations. Firstly, all of the macaques in the SIVART cohort received DTG/FTC/TDF. Since TDF can prevent weight gain [33,34], it might also have counteracted in part dolutegravir’s pro-adipogenic effect. Although our in vivo model recapitulated the pathophysiology of AT dysfunction in ART-controlled PLWH, only INSTIs were evaluated in our in vitro ASC model. We recently demonstrated that HIV/SIV infection per se has a specific influence on AT fibrosis and adipogenesis [3]. The combined or synergistic effects of INSTIs and HIV infection require further investigation, especially because HIV might influence brown adipogenesis [35]. Finally, in many of the experiments performed in vitro the strongest effect was observed in response to dolutegravir or bictegravir, while treatment of differentiating adipocytes with raltegravir caused very little or no significant effects. Thus, our data suggest that dolutegravir and bictegravir could exert a greater effect than raltegravir. Nevertheless, in vitro concentrations of active drugs could differ from the in vivo situation due to different protein levels in the medium and serum, thus explaining potential discrepancies.

## 5. Conclusions

In conclusion, we suggest that INSTIs can promote weight gain by reducing WAT-beiging capacities. We also propose that INSTI-induced adipocyte hypertrophy creates a hypoxic environment, which in turn promotes fibrosis and inhibits WAT beiging. By preventing energy expenditure, beiging inhibition would further favor lipid accumulation and thus, adipocyte hypertrophy resulting in insulin resistance.

## Figures and Tables

**Figure 1 cells-11-01841-f001:**
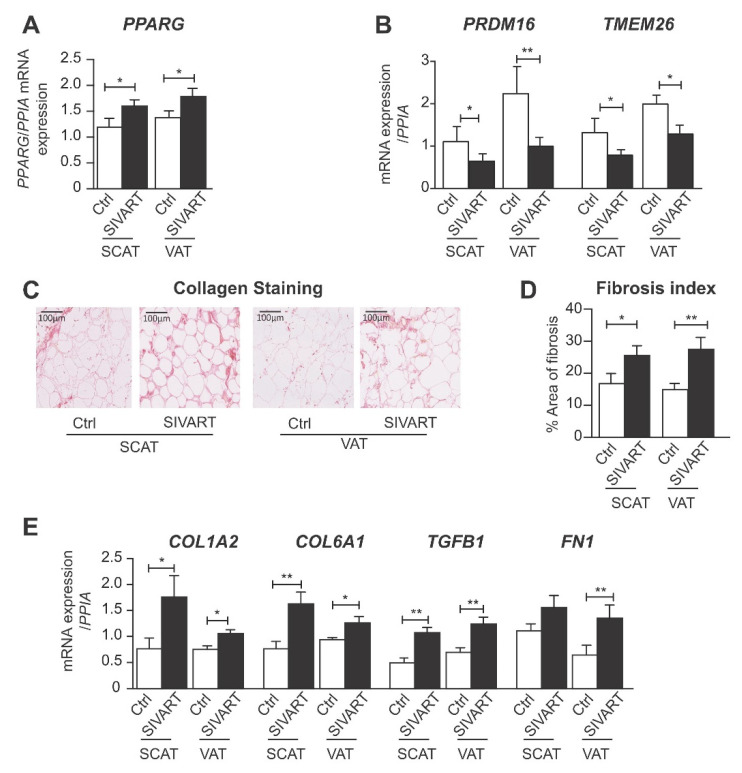
Subcutaneous and visceral adipose tissue from SIV-infected dolutegravir-treated macaques exhibit higher expression of PPARG and fibrosis-related genes and lower expression of beige adipocyte markers. Relative mRNA expression levels (in SCAT and VAT) of (**A**) peroxisome proliferator-activated receptor gamma (PPARG). (**B**) Positive-regulatory-domain-containing-16 (PRDM16) and transmembrane protein 26 (TMEM26) were normalized against that of peptidylprolyl isomerase A (*PPIA*). The results are expressed as the mean ± SEM PCR experiments performed in duplicates using SCAT and VAT from nine control uninfected and untreated macaques (Ctrl) and eleven SIV-infected and ART-treated macaques (SIVART). (**C**) Light microscopy analysis of adipose tissue depots stained with Sirius red to detect collagen fibers. Representative photographs are shown (scale bar, 100 μm). (**D**) The index of fibrosis was determined using a semi-automatic image analysis system in at least three randomly chosen regions, measured as described in the ‘Material and Methods’ section, using SCAT and VAT from seven control and ten SIVART macaques. (**E**) Relative mRNA expression levels of collagens, COL1A2, COL6A1, transforming growth factor beta 1 (TGFB1), and fibronectin 1 (FN1) normalized against that of *PPIA*. PCR experiments performed in duplicates using SCAT and VAT from nine control and eleven SIVART macaques. The results are expressed as the mean ± SEM. * *p* < 0.05, ** *p* < 0.01, vs. control animals.

**Figure 2 cells-11-01841-f002:**
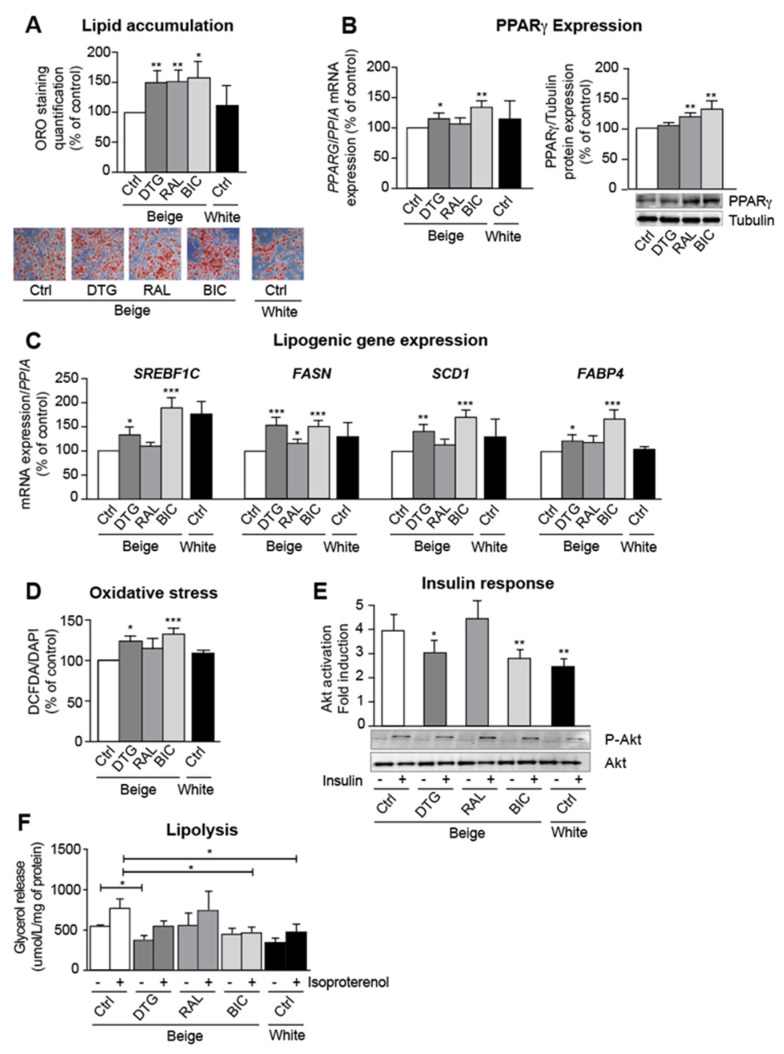
INSTI treatment is associated with greater lipid accumulation, pro-lipogenic marker expression, and insulin resistance in beige adipocytes in vitro. ASCs were differentiated into beige adipocytes in the presence of DMSO (Ctrl), dolutegravir/DTG, raltegravir/RAL, or bictegravir/BIC or differentiated into white adipocytes with DMSO (Ctrl) for 14 days. (**A**) Representative pictures are shown (Magnification X10). Oil-Red-O staining was expressed as the mean ± SEM % of control cells (*n* = 9, in triplicate). (**B**) The relative mRNA expression levels of PPARG were normalized against that of *PPIA*. The results are expressed as the mean ± SEM % of control cells (left panel) (*n* = 9, in duplicate). Whole-cell lysates were prepared and analyzed using immunoblots. Representative immunoblots of PPARγ and tubulin (loading control) are shown. Densitometry analyses (*n* = 6) against tubulin as loading control were performed and expressed as a mean % of the value for control cells ± SEM (right panel) (*n* = 6). (**C**) The relative mRNA expression levels of sterol-regulatory-element-binding-factor-1c (SREBF1C), fatty acid synthase (FASN), stearoyl-CoA-desaturase-1 (SCD1), and fatty-acid-binding-protein-4 (FABP4) were normalized against that of *PPIA*. The results are expressed as the mean ± SEM % of control cells (left panel) (*n* = 9, in duplicate). (**D**) ROS production was assessed by measuring the CM-H2DCFDA fluorescence (normalized against DAPI). The results are expressed as the mean ± SEM % of control cells (*n* = 9, in triplicate). (**E**) The cells were stimulated (or not) with 100 nM insulin for 7 min. Whole-cell lysates were prepared and analyzed using immunoblots. Representative immunoblots of phospho-Akt, total Akt are shown. The fold induction by insulin was quantified. The results are expressed as the mean ± SEM (*n* = 6). (**F**) Lipolysis was evaluated by measuring glycerol release in adipocytes stimulated (or not) with 1 μM isoproterenol (normalized against protein content). The results are expressed as the mean ± SEM (*n* = 3, in duplicate). * *p* < 0.05, ** *p* < 0.01, *** *p* < 0.001 vs. control cells.

**Figure 3 cells-11-01841-f003:**
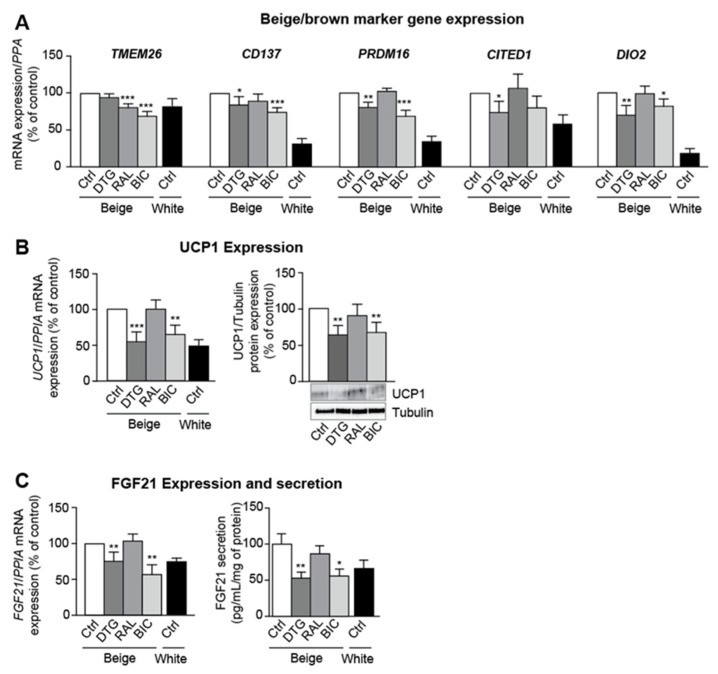
INSTI treatment during beige adipocyte differentiation is associated with the inhibition of beige adipogenesis. ASCs were differentiated into beige adipocytes in the presence of DMSO (Ctrl), dolutegravir/DTG, raltegravir/RAL, or bictegravir/BIC or differentiated into white adipocytes with DMSO (Ctrl) for 14 days. (**A**) Relative mRNA expression levels of TMEM26, cluster of differentiation 137 (CD137), PRDM16, Cbp/P300-interacting-transactivator-1 (CITED1), and iodothyronine-deiodinase-2 (DIO2), were normalized against that of *PPIA*. The results are expressed as the mean ± SEM % of control cells (*n* = 9, in duplicate). (**B**) Relative mRNA expression levels of uncoupling protein 1 (UCP1) was normalized against that of *PPIA* (left panel). The results are expressed as the mean ± SEM % of control cells (*n* = 9, in duplicate). Whole-cell lysates were prepared and analyzed using immunoblots. Representative immunoblots of UCP1 and tubulin are shown (right panel). Densitometry analyses against tubulin as loading control were performed and expressed as a mean % of the value for control cells ± SEM (*n* = 6). (**C**) Relative mRNA expression of fibroblast-growth factor-21 (FGF21) was normalized against that of *PPIA* (left panel). The results are expressed as the mean ± SEM % of control cells (*n* = 9, in duplicate). After 14 days of differentiation with INSTIs, the levels of FGF21 in the culture medium (right panel) from the last 48 h were determined with ELISA (*n* = 6, in duplicate). The results are expressed as the mean ± SEM. * *p* < 0.05, ** *p* < 0.01, *** *p* < 0.001 vs. control cells.

**Figure 4 cells-11-01841-f004:**
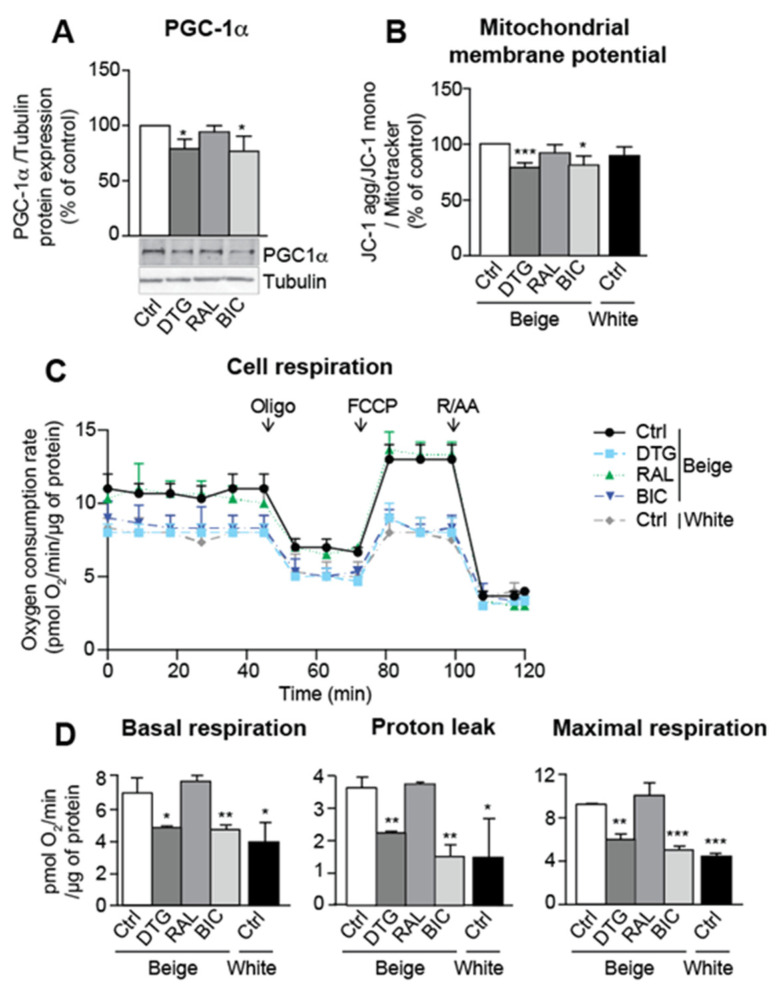
Treatment with an INSTI during beige adipocyte differentiation is associated with impaired mitochondria function and respiration. ASCs were differentiated into adipocytes as described in Figure 2. (**A**) Whole-cell lysates were prepared and analyzed using immunoblots. Representative immunoblots of PPARγ-coactivator-1α (PGC-1α) and tubulin are shown. Densitometry analyses (*n* = 6) against tubulin as loading control were performed and expressed as a mean % of the value for control cells ± SEM (*n* = 6). (**B**) The cationic dye, JC-1, was used to evaluate the mitochondrial membrane potential. The fluorescence results normalized against the mitochondrial mass (evaluated using MitoTracker Red dye) are expressed as the mean ± SEM % JC-1 aggregate/monomer ratio, relative to control cells (*n* = 9, in duplicate). (**C**) The oxygen consumption rate (OCR) of cells treated sequentially with oligomycin (Oligo), carbonyl-cyanide-p-trifluoromethoxyphenyl-hydrazon (FCCP), or rotenone and antimycin A (R/AA). OCR were normalized against the protein content and expressed as the mean ± SEM (*n* = 3, in quadruplicate). (**D**) Basal respiration, proton leakage, and maximal respiration were calculated, as described in Material and Methods. * *p* < 0.05, ** *p* < 0.01, *** *p* < 0.001 vs. control cells.

**Figure 5 cells-11-01841-f005:**
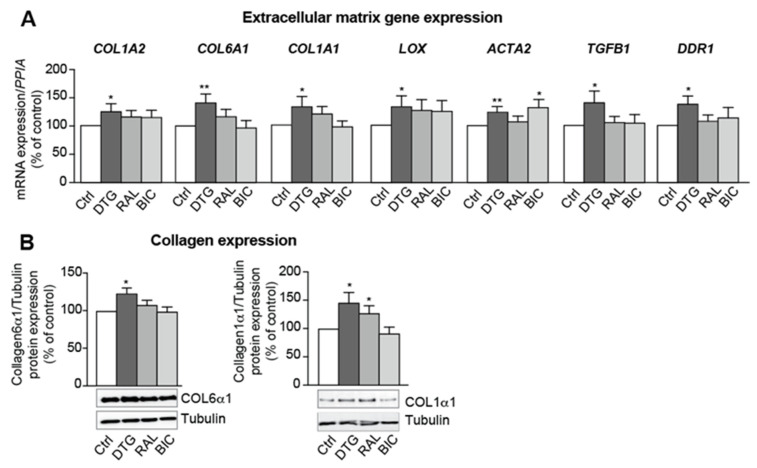
Treatment with INSTIs is associated with elevated extracellular matrix component expression in beige adipocytes. ASCs were differentiated into adipocytes, as described in Figure 2. (**A**) Relative mRNA expression levels of collagens (COL1A2, COL6A1, COL1A1), lysyl oxidase (LOX), actin alpha 2, smooth muscle (ACTA2), TGFB1, and DDR1, were normalized against that of *PPIA*. The results are expressed as the mean ± SEM % of control cells (*n* = 9, in duplicate). (**B**) Whole-cell lysates were prepared and analyzed using immunoblots. Representative immunoblots of COL6α1, COL1α1, and tubulin are shown. Densitometry analyses against tubulin as loading control were performed and expressed as a mean % of the value for control cells ± SEM (*n* = 6). * *p* < 0.05, ** *p* < 0.01 vs. control cells.

**Figure 6 cells-11-01841-f006:**
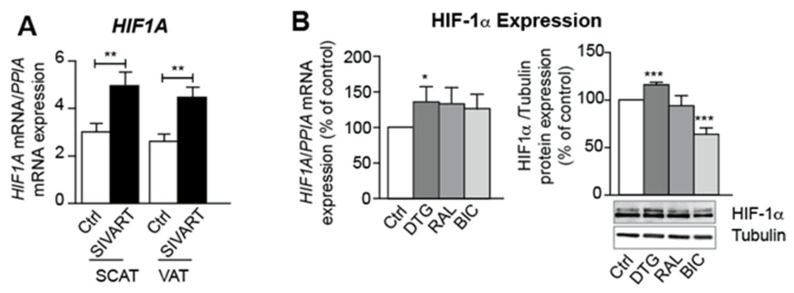
Elevated levels of *HIF1A* expression are observed in SCAT and VAT from INSTI-treated, SIV-infected macaques and in adipocytes derived from INSTI-treated, beige-differentiated ASCs. Relative mRNA expression levels of hypoxia-inducible-factor-1-alpha (*HIF1A*) normalized against that of *PPIA* in (**A**) SCAT and VAT from SIVART or control macaques. PCR experiments performed in duplicates using SCAT and VAT from nine control and eleven SIVART macaques. The results are expressed as the mean ± SEM. ASCs were differentiated into adipocytes, as described in Figure 2. (**B**) Relative mRNA expression *HIF1A* was normalized against that of *PPIA*. The results are expressed as the mean ± SEM % of control cells (*n* = 9, in duplicate) (left panel). Whole-cell lysates were prepared and analyzed using immunoblots. Representative immunoblots of HIF-1α and tubulin are shown. Densitometry analyses against tubulin as loading control were performed and expressed as a mean % of the value for control cells ± SEM (*n* = 6) (right panel). * *p* < 0.05, ** *p* < 0.01, *** *p* < 0.001 vs. control animals or vs. control cells.

**Figure 7 cells-11-01841-f007:**
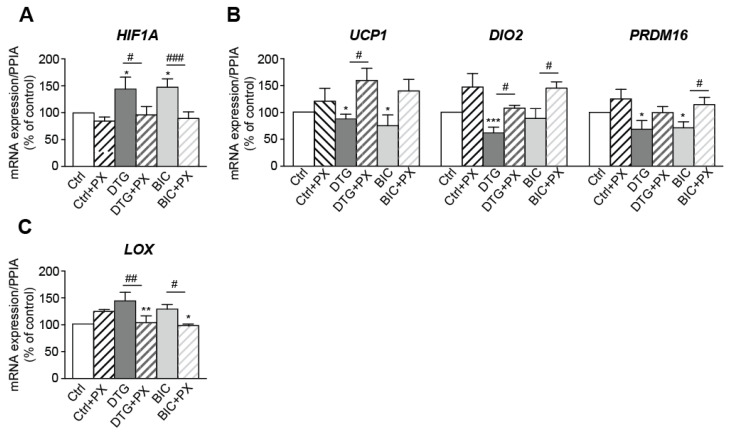
The inhibition of *HIF1A* by PX478 prevented the deleterious effects of INSTIs on beiging and fibrosis. ASCs were concomitantly treated with INSTIs and PX478 (HIF-1α inhibitor) throughout the differentiation process. The relative mRNA expression levels of (**A**) *HIF1A*, (**B**) *UCP1, DIO2; PRDM16*, and (**C**) *LOX* were normalized against that of *PPIA.* The results are expressed as the mean ± SEM % of control cells (*n* = 3, in duplicate). * *p* < 0.05, ** *p* < 0.01, *** *p* < 0.001 vs. control cells. ^#^ *p* < 0.05, ^##^ *p* < 0.01^, ###^ *p* < 0.001 PX478-treated vs. non treated cells.

**Table 1 cells-11-01841-t001:** Correlations (Pearson’s test) between *HIF1A* and beiging-, fibrosis- and lipid-related markers in vivo. Correlation between *HIF1A* gene expression and beiging-, fibrosis-, and lipid-related gene expression in SCAT and VAT from SIVART and control macaques. Correlations were determined using PCR data obtained using SCAT and VAT from six control uninfected and untreated macaques (Ctrl) and eight SIV-infected and ART-treated macaques (SIVART).

		*HIF1A*			
		SCAT		VAT	
		r	*p*	r	*p*
Beiging	*PRDM16*	−0.665	0.018	−0.188	0.251
Fibrosis	*COL6A1*	0.468	0.054	0.655	0.003
	*COL1A2*	0.021	0.475	0.527	0.027
	*TGFB1*	0.617	0.012	0.744	<0.001
Adipogenesis	*PPARG*	0.622	0.015	0.073	0.395

**Table 2 cells-11-01841-t002:** Correlations (Pearson’s test) between *HIF1A* and beiging, fibrosis and lipid accumulation in vitro. Correlation between *HIF1A* gene expression and beiging-, fibrosis-, and lipid-related gene expression or lipid accumulation in INSTI-treated beige adipocytes. Correlations were determined using PCR data obtained using ASCs (*n* = 9).

		*HIF1A*	
		r	*p*
Beiging	*PRDM16*	−0.410	0.009
	*UCP1*	−0.435	0.008
	*DIO2*	−0.663	<0.001
Fibrosis	*COL6A1*	−0.040	0.412
	*COL1A1*	0.286	0.059
	*COL1A2*	0.559	<0.001
	*LOX*	0.606	<0.001
	*ACTA2*	0.561	<0.001
	*TGFB1*	0.123	0.250
	*DDR1*	0.480	0.002
Lipid metabolism	*PPARG*	0.05	0.397
	*FABP4*	0.438	0.006
	Oil-Red-O	0.357	0.043

## Data Availability

Not applicable.

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
