# Peer review of "Inhibition of Adipose Tissue Beiging by HIV Integrase Inhibitors, Dolutegravir and Bictegravir, Is Associated with Adipocyte Hypertrophy, Hypoxia, Elevated Fibrosis, and Insulin Resistance in Simian Adipose Tissue and Human Adipocytes"

_cells, 2022, doi:10.3390/cells11111841_

Round 1
Reviewer 1 Report
The manuscript “Inhibition of adipose tissue beiging by HIV integrase inhibitors…” authored by K.N. Ayissi et al. is devoted to the problem of weight gain and fat expansion in HIV-infected ART-control people living with HIV who switch to the integrase-strand-transfer-inhibitors (INSTIs) dolutegravir and bictegravir. The authors hypothesized that INSTIs could alter adipose tissue (AT) by inhibition of the beiging process. In order to confirm their hypothesis they studied expression of beige markers and cell respiration in human stromal cells exposed to INSTIs in vitro, and AT from SIVART macaques. The obtained data confirm that indeed INSTIs promoted lipid accumulation, inhibited lipolysis, decrease WAT beiging capacity and favor adipocyte hypertrophy.
The manuscript is well written, the methodology is adequate, tables and figures are clear. Overall, the paper is of high quality.
Specifically, the line 333 sentence: “The AT expansion associated with obesity results in adipocyte hypertrophy”… is somewhat unclear – it appears that obesity and adipocyte hypertrophy are both cause and effect. Usually, it is said that adipose tissue expands due to hypertrophy and hyperplasia of adipocytes. Hyperplasia is not mentioned at all in the manuscript, it is probably worthwhile to add couple sentences about it in “Discussion”.
One more concern is line 374 – Informed Consent Statement, which says “Not applicable”, although informed consent was discussed in “Materials and Methods”, lines 100 – 103.
Finally, the role of browning of WAT in tumor progression is currently being studied (see e.g. Alvarez-Artime etal., Int. J. Mol. Sci. 2021, 22, 5560), and relationship between BAT activation and tumor progression has been established in relation to cachexia, evidently the reverse process. This may be interesting for “Discussion”.
The above specific points in no way undermine the high quality of the reviewed manuscript.
Author Response
Comments and Suggestions for Authors
The manuscript “Inhibition of adipose tissue beiging by HIV integrase inhibitors…” authored by K.N. Ayissi et al. is devoted to the problem of weight gain and fat expansion in HIV-infected ART-control people living with HIV who switch to the integrase-strand-transfer-inhibitors (INSTIs) dolutegravir and bictegravir. The authors hypothesized that INSTIs could alter adipose tissue (AT) by inhibition of the beiging process. In order to confirm their hypothesis, they studied expression of beige markers and cell respiration in human stromal cells exposed to INSTIs in vitro, and AT from SIVART macaques. The obtained data confirm that indeed INSTIs promoted lipid accumulation, inhibited lipolysis, decrease WAT beiging capacity and favor adipocyte hypertrophy.
The manuscript is well written, the methodology is adequate, tables and figures are clear. Overall, the paper is of high quality.
Authors response: We would like to thank the reviewer for his/her very positive comments regarding our manuscript.
Further comments
Point 1: Specifically, the line 333 sentence: “The AT expansion associated with obesity results in adipocyte hypertrophy”… is somewhat unclear – it appears that obesity and adipocyte hypertrophy are both cause and effect. Usually, it is said that adipose tissue expands due to hypertrophy and hyperplasia of adipocytes. Hyperplasia is not mentioned at all in the manuscript, it is probably worthwhile to add couple sentences about it in “Discussion”.
Authors response: We agree with the reviewer’s comment and as requested we have clarified the point about adipose tissue expansion including hyperplasia and hypertrophy in the introduction (line 60).and discussion sections (line 415).
Point 2: One more concern is line 374 – Informed Consent Statement, which says “Not applicable”, although informed consent was discussed in “Materials and Methods”, lines 100 – 103.
Authors response: We made the changes accordingly. Informed Consent Statement form was added (line 503).
Point 3: Finally, the role of browning of WAT in tumor progression is currently being studied (see e.g. Alvarez-Artime et al., Int. J. Mol. Sci. 2021, 22, 5560), and relationship between BAT activation and tumor progression has been established in relation to cachexia, evidently the reverse process. This may be interesting for “Discussion”.
Authors response: As suggested by Reviewer, we added a paragraph in the Discussion section on the relationship between WAT beiging and cancer (line 434) and on the incidence of cancers in HIV-infected patients receiving an INSTI and added the corresponding references.
The above specific points in no way undermine the high quality of the reviewed manuscript.
Authors response: Thank you.
Reviewer 2 Report
In this study, the authors investigate the effect of integrase HIV inhibitors (INSTIs), including DTG and BIC, in two model systems, subcutaneous and visceral adipose tissue (SCAT & VAT, respectively) from SIV-infected cynomolgus macaques, or in vitro differentiated human adipocytes. Given the recent concerns about excess weight gain among users of certain ARVs (DTC, BIC, TAF), this study is important and timely.
Previous studies from this group demonstrated changes in fat metabolism in untreated SIV-infected macaques, and separately demonstrated increased adipogenesis, adipose tissue fibrosis, oxidative stress, mitochondrial dysfunction, and insulin resistance in uninfected macaques treated with DTG/TDF/FTC. In the current study, they demonstrate that DTG and another second generation INSTI, BIC, are associated with lower levels of beige markers and adipocyte properties (increased lipogenic gene expression, oxidative stress/ROS production and insulin resistance) compared to controls. The study therefore makes an important contribution towards understanding how these INSTIs affect adipose tissue.
Although the authors shown many associations (reduced expression of beige markers, increased fibrosis, ROS production, insulin resistance and increased expression of HIF1α) with INSTI use which may point to a mechanism for how the DTG or BIC lead to weight gain, these correlations does not immediately demonstrate causation. The working hypothesis is that INSTIs alter adipose tissue via beiging inhibition. The authors observe increased expression of HIF1α, and therefore assume a mechanism where INSTIs cause hypoxia and suggest that this is the cause of the adipose tissue dysregulation. However since all experiments are done at ‘steady state’ (macaques were treated with DTG for >2 years, and the AT stem cells differentiated in the presence of drug cells for 14 days), it is difficult to define at which step, or by what mechanism of action, INSTIs increase adipose tissue dysfunction. This uncertainty might be better reflected in the Abstract and Discussion sections.
Further comments:
- A key part of the argument that DTG and BIC inhibit AT beiging is based on correlation between the expression of various gene markers using Pearson correlation. Please can you show the scatter plots on which the calculations of the regression coefficients (Table 1 and Table 2) are based (at least in a supplementary document like the one showing the gels and blots). Pearson correlation should only be calculated where the relationship is approximately linear, and visualisation of the scatter of the sample points is helpful to understand the extent of the correlation.
- Please also provide more detail of how the means and standard errors were calculated – number of experimental replicates, at which point ratios between the gene of interest and PPIA were calculated, and the precise number of samples included in the mean and SE calculation.
- The macaques have been treated with DTG/TDF/FTC for more than 2 years. Given the effect of SIV infection alone on adipose tissue, could you report whether and how long the animals had a suppressed viral load and whether virus or viral proteins persisted in the adipose tissue biopsies.
- Minor points:
- Please clarify the control macaques – I presume these are SIV-negative and not treated with ARVs. This could be specified explicitly.
- Given the previous study, a summary comparing uninfected untreated macaques with SIV infected animals (ARV-treated or untreated) and uninfected INSTI-treated animals, a summary of all these results might be included, showing important differences between these groups.
- Are the macaque adipose samples obtained by biopsy (line 78) or necropsy (line 95)? Please clarify.
- Please specify which p-values are based on Student’s t-test and which on the non-parametric test. Also it would help to report n for each comparison.
- Please provide scale bars for the images in Fig. 1 and Fig. 2 as the magnification depends on size of the images when they are reproduced.
- Consistent use of abbreviations - PPARγ (line 112) or PPARG (elsewhere) and e.g. this varies between e.g. PPARG in Fig. 1 and PPARγ in Fig 2B, or both in line 197.
- More explicit that for many assays the strongest effect was with BIC, while treatment of differentiating adipocytes with RAL caused only small/no significant effects.
Author Response
Comments and Suggestions for Authors
Although the authors shown many associations (reduced expression of beige markers, increased fibrosis, ROS production, insulin resistance and increased expression of HIF1α) with INSTI use which may point to a mechanism for how the DTG or BIC lead to weight gain, these correlations does not immediately demonstrate causation. The working hypothesis is that INSTIs alter adipose tissue via beiging inhibition. The authors observe increased expression of HIF1α, and therefore assume a mechanism where INSTIs cause hypoxia and suggest that this is the cause of the adipose tissue dysregulation. However since all experiments are done at ‘steady state’ (macaques were treated with DTG for >2 years, and the AT stem cells differentiated in the presence of drug cells for 14 days), it is difficult to define at which step, or by what mechanism of action, INSTIs increase adipose tissue dysfunction. This uncertainty might be better reflected in the Abstract and Discussion sections.
Authors response: We would like to thank the Reviewer for the careful reading of the manuscript. We agree with reviewer’s comment regarding the role of hypoxia. Therefore, we carried out some new experiments to further establish the role of HIF-1a, using a potent inhibitor (PX478). We observed that PX478 prevented the INSTI-induced higher HIF1A gene expression. Accordingly, PX478-treatment prevented the DTG and BIC-induced decreased level of beige markers and increased the level of the fibrosis marker gene LOX. Therefore, these results indicate that HIF-1a is an initial step in the beige inhibiting effect of INSTIs. These new results are given in the results section and presented as a novel figure 7. We have not modified the abstract since we performed new experiments showing the causative link between higher HIF1α and beiging inhibition. Nonetheless changes were made in the material and methods section (line 124 and line 178) and commented in the results (line 332) and discussion sections (line 428).
Further comments:
Point 1: A key part of the argument that DTG and BIC inhibit AT beiging is based on correlation between the expression of various gene markers using Pearson correlation. Please can you show the scatter plots on which the calculations of the regression coefficients (Table 1 and Table 2) are based (at least in a supplementary document like the one showing the gels and blots). Pearson correlation should only be calculated where the relationship is approximately linear, and visualisation of the scatter of the sample points is helpful to understand the extent of the correlation.
Author response: As requested, we added a supplemental figure 2 displaying the scatter plots along with the correlations. Since the relationships are approximately linear, this validates the use of the Pearson correlation.
Point 2: Please also provide more detail of how the means and standard errors were calculated – number of experimental replicates, at which point ratios between the gene of interest and PPIA were calculated, and the precise number of samples included in the mean and SE calculation.
Author response: We have included the number of experiment replicates in all figures. Moreover, in order to explicit how means and standard errors of the means were determined we added this sentence in the material and methods section (line 146) : « PCR raw data were analyzed using relative quantification approach with efficiency correction. The internal reference gene, PPIA, was measured in addition to the target genes for each sample. Ratios between the gene of interest and the reference gene were evaluated and compared. »
Point 3: The macaques have been treated with DTG/TDF/FTC for more than 2 years. Given the effect of SIV infection alone on adipose tissue, could you report whether and how long the animals had a suppressed viral load and whether virus or viral proteins persisted in the adipose tissue biopsies.
Author response: Animals were treated during the chronic phase of infection (at day 120 post-infection). At that time point, the median of viral load was 1.0 104 [IQR 1.0 104-5.9 104]. In all animals, treatment initiation resulted in a drastic reduction of viral load, detectable as soon as day 7 post-treatment (median: 52 copies/mL [IQR 11-120]). From week 6 post treatment initiation, viral load was maintained below 50 copies/mL until the sacrifice of the animals when adipose tissue samples were withdrawn. We added this point in the Material and Methods section (lines 96 and 100). Analyses of viral load in tissues are not available for these animals.
Minor points:
- Please clarify the control macaques – I presume these are SIV-negative and not treated with ARVs. This could be specified explicitly.
Author response: As presumed by the reviewer, the control macaques were indeed uninfected and did not receive any treatment. This point was clarified in the Methods section (line 99).
- Given the previous study, a summary comparing uninfected untreated macaques with SIV infected animals (ARV-treated or untreated) and uninfected INSTI-treated animals, a summary of all these results might be included, showing important differences between these groups.
Author response: As requested, we have included, as a supplemental table 3, a small summary of the impact of SIV, ART or both on PPARG, collagen gene expression, and fibrosis index. This Table is now mentioned in the discussion section (lines 378 and 449).
- Are the macaque adipose samples obtained by biopsy (line 78) or necropsy (line 95)? Please clarify.
Author response: Adipose tissue samples were obtained by necropsy. The text in introduction section has been changed accordingly.
- Please specify which p-values are based on Student’s t-test and which on the non-parametric test. Also it would help to report n for each comparison.
Author response: To simplify the analysis we have decided to perform all statistical analysis using non parametric tests and changed the Material and Methods section accordingly. We have included the number of experiment replicates in all figures. The n is the same for all conditions. We also added statistical analysis with a two-way ANOVA to evaluate the impact of PX478 (HIF1alpha inhibitor) on gene expression in ASCs.
- Please provide scale bars for the images in Fig. 1 and Fig. 2 as the magnification depends on size of the images when they are reproduced.
Author response: As requested we added scale bars on Figure1. However, we did not add scale bars on Figure 2 as all pictures were taken at the magnification without any modification.
- Consistent use of abbreviations - PPARγ (line 112) or PPARG (elsewhere) and e.g. this varies between e.g. PPARG in Fig. 1 and PPARγ in Fig 2B, or both in line 197.
Author response: We followed the instruction to authors, therefore the guidelines for formatting gene and protein names. For Humans and non-human primates: Gene symbols contain italicized characters that are all in upper-case, thus PPARG. Protein symbols are identical to their corresponding gene symbols except that they are not italicized, and can contain greek symbols, thus PPARγ.
- More explicit that for many assays the strongest effect was with BIC, while treatment of differentiating adipocytes with RAL caused only small/no significant effects.
Author response: In line with our present data, we observed in a previous study that raltegravir (RAL) treatment was associated with a slight elevation in lipid accumulation into adipocytes but did not affect adipogenesis per se, as the expression of the pro-adipogenic markers PPARG was not elevated as compared to the control. Thus, our data suggest that dolutegravir (DTG) and bictegravir (BIC) could exert a greater effect than RAL. Nevertheless, in vitro concentrations of active drugs could differ from the in vivo situation due a different protein level in the culture medium and in serum. Changes in the discussion were made accordingly (line 451).
Round 2
Reviewer 1 Report
All comments have been adquately addressed
Reviewer 2 Report
Thank you for the opportunity to review this very interesting and timely manuscript.
All my comments have been addressed